# Identification of β-Glucosidase Activity of *Lentilactobacillus buchneri* URN103L and Its Potential to Convert Ginsenoside Rb1 from *Panax ginseng*

**DOI:** 10.3390/foods11040529

**Published:** 2022-02-12

**Authors:** Gereltuya Renchinkhand, Urgamal Magsar, Hyoung Churl Bae, Suk-Ho Choi, Myoung Soo Nam

**Affiliations:** 1Division of Animal Resource Science, Chungnam National University, Daejeon 34134, Korea; handgai@yahoo.com (G.R.); magsarurgamal@gmail.com (U.M.); hcbae@cnu.ac.kr (H.C.B.); 2Department of Animal Biotechnology, Sangji University, Wonju 26339, Korea; shchoi@sangji.ac.kr

**Keywords:** β-glucosidase, ginsenoside Rb1, Rd, Rg3, hydrolyze, *Lentilactobacillus buchneri* URN103L

## Abstract

*Lentilactobacillus buchneri* isolated from Korean fermented plant foods produces β-glucosidase, which can hydrolyze ginsenoside Rb1 from Panax ginseng to yield ginsenoside Rd. The aim of this study was to determine the mechanisms underlying the extracellular β-glucosidase activity obtained from *Lentilactobacillus buchneri* URN103L. Among the 17 types of lactic acid bacteria showing positive β-glucosidase activity in the esculin iron agar test, only URN103L was found to exhibit high hydrolytic activity on ginsenoside Rb1. The strain showed 99% homology with *Lentilactobacillus buchneri* NRRLB 30929, whereby it was named *Lentilactobacillus buchneri* URN103L. Supernatants of selected cultures with β-glucosidase activity were examined for hydrolysis of the major ginsenoside Rb1 at 40 °C, pH 5.0. Furthermore, the β-glucosidase activity of this strain showed a distinct ability to hydrolyze major ginsenoside Rb1 into minor ginsenosides Rd and Rg3. *Lentilactobacillus buchneri* URN103L showed higher leucine arylamidase, valine arylamidase, α-galactosidass, β–galactosidase, and β-glucosidase activities than any other strain. We conclude that β-glucosidase from *Lentilactobacillus buchneri* URN103L can effectively hydrolyze ginsenoside Rb1 into Rd and Rg3. The converted ginsenoside can be used in functional foods, yogurts, beverage products, cosmetics, and other health products.

## 1. Introduction

Korean ginseng (*Panax ginseng* C.A. Mey) contains approximately 200 substances, including ginsenosides, polysaccharides, polyacetylenes, peptides, and amino acids. To date, 100 ginsenosides have been isolated, with Rb1, Rb2, Rc, Re, and Rg1 constituting more than 80% of all ginsenosides reported [1,2]. The most abundant components of Korean ginseng are ginsenosides (ginseng saponin) and polysaccharides, which are not absorbed by the human intestines due to their chemical hydrophilicity. However, if these constituents come into contact with the intestinal microbiota in the digestive tract, microbes use them to produce minor metabolites, such as compound K and protopanaxatriol, which are easily absorbed by the human intestines. These metabolites exhibit important pharmacological functions, including antitumor, antidiabetic, anti-inflammatory, anti-allergic, immunomodulatory, and neuroprotective activities [3,4,5]. Lactic acid bacteria (LAB) are an integral part of traditional food processing and preservation technologies, including the fermentation of dairy products, plants, and meat. The hydrolysis of plant metabolite glucoconjugates by the β-glucosidase activities of LAB is a significant contribution to the biological activity and dietary attributes of fermented food [6].

Furthermore, LAB exhibiting extracellular β-glucosidase activity are present in fermented plant foods; indeed, certain LAB strains possessing β-glucosidase activity may have the potential to metabolize ginsenoside Rb1 from *P. ginseng.* The hydrolysis of ginsenosides to ginsenosides Rd and Rg3 by various bacteria can increase their beneficial effects. Therefore, research has been conducted to identify microorganisms that can efficiently metabolize major ginsenosides, e.g., Rb1 or Re, to minor ginsenosides, e.g., Rd, F2, compound K, or Rh2 [7]. In addition, ginsenoside Rd can enhance the proliferation of neural stem cells in vivo and in vitro [8] and inhibit the proliferation and survival of gastric and breast cancer cells by influencing the transient receptor potential melastatin 7 (TRPM7) channel [9]. Thus, recently, ginsenoside Rg3 was tested as a new anticancer drug for therapy, in combination with chemotherapy, against lung, gastric, and esophageal cancer in China [10]. *Lentilactobacillus buchneri*, named in honor of the German bacteriologist E. Buchner, is an obligate heterofermentative facultative anaerobe, described as having diverse effects, such as the prevention of silage spoilage by yeast and molds [11]. Furthermore, buchnericin production by *L. buchneri* inhibits the growth of some species of *Listeria, Bacillus, Micrococcus, Enterococcus,* and *Vibrio* [12]. However, to date, there are no reports on the potential use of the extracellular β-glucosidase activity of *L. buchneri.* Therefore, the aim of this study was to determine the extent of ginsenoside Rb1 metabolism to Rd and Rg3 by the extracellular β-glucosidase activity obtained from *L. buchneri* URN103L.

## 2. Materials and Methods

### 2.1. Isolation of LAB with β-Glucosidase Activity from Fermented Plants

Lactobacillus strains were isolated from eight types of traditional Korean fermented plant foods randomly collected from the Daejeon area. Isolation of LAB with β-glucosidase activity from fermented plants was performed according to the method in [13]. The culture supernatants were filtered using a 0.2 μm syringe filter (HYUNDAT MICRO Co. LTD., Seoul, Korea) and mixed with 1 mg mL^−1^ standard saponin (Rg1 (4.80%), Re (12.95%), Rc (11.12%), Rd (5.87%), Rb2 (8.32%), Rb1 (15.91%)) in a 1:1 (*v*/*v*) ratio. The mixture of supernatant and standard saponin was incubated at 35 °C for 7 days at 190 rpm in a shaking incubator (Vision Scientific, Korea). The hydrolysis of saponin was monitored by thin-layer chromatography (TLC) analysis.

### 2.2. 16S rDNA Sequencing of Strain URN103L

16S rDNA sequencing of strain URN103L was performed according to the method in [13]. The primers used were 27F:5′-AGAGTTTGATCACTGGCTCAG-3′ and 1492R: 5′-GGTTACCTTGTTACGACTT-3′. PCR was performed using the 2 × PCR pre Mix (EF-taq) in a GeneAmp PCR system 2700 (Applied Biosystems, Singapore). The NCBI GenBank registration number of strain URN103L is OM438138.1.

### 2.3. Identification of Enzyme Activity of the Strain URN103L Using the API ZYM Kit

Enzyme activity patterns of the selected strains were investigated using the API ZYM Kit for the research of enzymatic activity (bioMerieux, Marcy-I’Etoile, France), following manufacturer instructions [14].

### 2.4. Determination of Optimum pH for β-Glucosidase Hydrolysis of Ginsenoside Rb1

*L. buchneri* URN103L was inoculated in MRS broth at 37 °C for 24 h, after which the supernatant was collected by centrifugation at 7000× *g* for 15 min at 4 °C. The crude enzyme extract of the selected cultures was tested for the hydrolysis of major ginsenoside Rb1 (Daedook Bio, Korea) by β-glucosidase. The reaction mixture was prepared in a 1:1 (*v*/*v*) ratio of supernatant and 0.2 mg mL^−1^ of ginsenoside Rb1 (5 mM of sodium phosphate buffer) and filtered through a 0.2 μm syringe filter (HYUNDAT MICRO Co. LTD, South Korea). The reaction mixture was incubated at 35 °C for 7 days. β-glucosidase activity of the supernatant was conducted according to the method in [15].

### 2.5. Determination of Optimum Temperature for β-Glucosidase Hydrolysis of Ginsenoside Rb1

The optimum temperature for the hydrolysis of ginsenoside Rb1 by the crude enzyme extract of *L. buchneri* URN103L was determined at pH 7.0 at 30, 35, and 40 °C, over 7 days. Then, samples were extracted with butanol (ratio of sample: butanol, 1:1, v/v) prior to their concentration using a freeze-drying machine (IlshinBioBase, Yangjoo, Korea). Finally, the ginsenoside concentration was adjusted to 1 mg mL^−1^ for TLC analysis.

### 2.6. Thin-Layer Chromatography and High-Performance Liquid Chromatography (HPLC)

Fermentation characteristics of ginsenosides fermented by the strain *L. buchneri* URN103L were analyzed by TLC and HPLC, according to the method in [13].

### 2.7. Hydrolysis of Ginsenoside Rb1 by L. buchneri URN103L under Optimum Conditions

The hydrolysis of ginsenoside Rb1 was performed under the optimum conditions determined for β-glucosidase activity as described above. *L. buchneri* URN103L was inoculated in MRS broth (pH 7.0), and the crude enzyme extract was collected by centrifugation at 7000× *g* for 15 min at 4 °C. The reaction mixture in ratio 1:1 (*v*/*v*) of crude enzyme extract (pH 5.0) and 0.2 mg mL^−1^ of ginsenoside Rb1 (5 mM of sodium phosphate buffer, pH 5.0) was filtered through a 0.2 μm syringe filter and incubated at 35 °C for 3, 7, 10, and 14 days. Samples were concentrated using a freeze-drying machine (Ilshin Lab Co., ltd., Korea). Finally, the ginsenoside concentration was adjusted to 1 mg mL^−1^ for TLC and HPLC.

### 2.8. Fermentation Characteristics of a 20% Ginseng Root Solution Fermented by L. buchneri URN103L

Ginseng roots were sliced using a mixer for 5 min, diluted to 20%, and sterilized at 75 °C for 10 min. The resultant 20% ginseng root solution was incubated with 3% strain URN103L at 35 °C for 14 days. After 3, 7, and 14 days, viable cell counts were determined using BCP agar incubated at 37 °C for 48 h. Fermentation characteristics of ginsenosides fermented by the strain *L. buchneri* URN103L were analyzed by TLC and HPLC, according to the method in [13].

### 2.9. Statistics Analysis

All experiments were repeated in triplicates. One-way analysis of variance (ANOVA) and Duncan’s multiple range tests were conducted using SAS (Statistical Analysis System Institute, Version 9.4, Cary, NC, USA) to measure significant differences (*p* < 0.05). Data are presented as mean ± standard error.

## 3. Results and Discussion

### 3.1. Isolation and Screening of LAB with β-Glucosidase Activity from Fermented Plants

Thirty-seven types of pure culture of bacteria from fermented plants were selected in this study. The characteristics of ginsenoside hydrolysis and the utilization of esculin agar for the β-glucosidase production ability at 37 °C, over 72 h, were carefully investigated. Seventeen strains (61L, 63L, 64L, 65L, 66L, 67L, 68L, 70L, 71L, 72L, 72bS, 73sS, 73bS, 95L, 103L, 122, and 123) among the fermented plant isolates showed an esculin-positive reaction. Furthermore, among these esculin-positive colonies, eight (63L, 65L, 71L, 72S, 73S, 95L, 103L, and 123) showed hydrolysis of ginsenoside Rb1 and were analyzed for 16S rDNA sequencing. Isolated single colonies of strain 103L in BCP and esculin agar are shown in Figure 1. Many researchers have isolated *L. buchneri* with various biological activities and probiotic properties [16,17]. Indeed, the isolated *L. buchneri* KU200793 has high γ-aminobutyric acid (GABA) production ability, and the GABA produced by the isolate was found to have a neuroprotective function and to protect MPP+-stressed SH-SY5Y cells more effectively than other LAB [16]. Further, *L. buchneri* KU200793 can reportedly be used as a probiotic starter in functional foods to prevent Parkinson’s disease [17]. Additionally, *L. buchneri* P2 isolated from pickled juice has shown probiotic properties, such as cholesterol reduction, bile tolerance, and antimicrobial activity [18]. In this study, we isolated *L. buchneri* URN103L with β-glucosidase activity that can hydrolyze ginsenoside from ginseng obtained from fermented plants.

### 3.2. Selection of Colonies That Can Hydrolyze Ginsenosides

Among the selected colonies, strain URN103L actively metabolized major ginsenoside Rb1 to minor ginsenoside Rd. However, this strain did not metabolize ginsenoside Rb1 to ginsenosides Rb2 and Re. Finally, TLC analysis (Figure 2) showed that LAB showing extracellular β-glucosidase activity only weakly metabolized ginsenoside Rb1 to ginsenoside Rd [13,14].

### 3.3. β-Glucosidase Activity of the Selected Strain URN103L

API ZYM is a simple, rapid system for the detection of bacterial enzymes that allows the detection of enzyme activity on 19 substrates within 4 h [19]. Among the isolates obtained here, strain URN103L showed the highest level of β-glucosidase activity. Strain URN103L showed five types of enzymatic activities related to carbohydrate hydrolysis, namely, α-galactosidase (4), β-galactosidase (5), β-glucuronidase (2), α-glucosidase (3), and β-glucosidase (5). Among LAB isolated from kimchi, *L. rhamnosus* GG and *L. buchneri* KU20073 showed the highest level of β-glucosidase, α-glucosidase, and β-galactosidase activities, which are important enzymes for the hydrolysis of glycosidic linkages [18]. Strain URN103L produced four types of proteolytic enzymes, and its activity was similar to that of other LAB. Among these proteolytic enzymes, leucine arylamidase and valine arylamidase showed high activity in the isolated strain URN103L. *L. buchneri* KU20073 produces leucine arylamidase and valine arylamidase with the highest activity, similar to isolated strain URN103L [17]. Many types of bacteria have been isolated from fermented plants, and their intracellular and extracellular β-glucosidase activities have been identified [13]. Esculin is a substrate used to quickly screen and identify bacteria with β-glucosidase activity and is used to add ferrium to bacterial media [20]. Many researchers have studied the β-glucosidase activity of microorganisms, including *L. casei* subsp. *rhamnosus* [21], *Leuconostos mesenteroides* [22], and *Lactobacillus plantarum* [23], among LAB species.

### 3.4. 16S rDNA Sequencing of Strain URN103L

Based on the analysis of the 16S rDNA sequences, the *L. buchneri* group is located within the family Lactobacillaceae. This group contained only obligate heterofermentative Lactobacilli. The selected eight types of colonies with an esculin-positive reaction were analyzed by 16S rDNA sequencing to determine which species would be matched with strain URN103L, which was selected as the LAB strain among the 14 isolated strains with β-glucosidase activity. The DNA sequence of strain URN103L was 1474 bp, which was compared with the sequences in the NCBI database using BLAST. Phylogenetic trees of strain URN103L were downloaded from the NCBI database and modified in Adobe Illustrator (Figure 3), and they were found to be identical to *L. buchneri* sp. Indeed, DNA analysis revealed that URN103L showed 99% homology with the nucleotide sequence of the identified *L. buchneri*. Therefore, strain URN103L was named *L. buchneri* URN103L.

### 3.5. Optimum pH and Temperature of β-Glucosidase and Conversion of Ginsenoside Rb1

The optimum pH and temperature conditions for the activity of the crude enzyme extract were determined to attain the optimum hydrolytic activity for the conversion of ginsenoside Rb1 to Rd. The optimum pH for the crude β-glucosidase extract from *L. buchneri* URN103L to hydrolyze ginsenoside Rb1 was found to be 5.0 (Figure 4). Different values of the optimum pH have been reported for different species of LAB, such as *L. rhamnosus* NRRL B442 [24] and *L. pentosus* strain 6105 [25], for which the corresponding optimum pH was found to be 6.4 and 7.0, respectively. Similarly, to determine the optimum temperature, the hydrolysis temperature of ginsenoside Rb1 by the crude enzyme extract was adjusted to 30, 35, and 40 °C, at pH 7.0. As shown in Figure 5, 35 °C was found to be the optimum temperature for the crude β-glucosidase extract. Based on TLC data analysis (Figure 4 and Figure 5), the optimum conversion conditions were determined to be pH 5 and 35 °C. Ginsenoside Rb1, which had the weakest band density at pH 5 and 35 °C, was converted to Rd and Rg3. The optimum temperature for the β-glucosidase activity of *L. rhamnosus* NRRL B442 [26] was reportedly 46 °C. Meanwhile, the optimum temperature of β-glucosidase isolated from *L. pentosus* strain 6105 was determined to be 37 °C, using the conversion of ginsenoside Rb1 to ginsenoside Rd, similar to *L. buchneri* URN103L [27].

### 3.6. Metabolism of Ginsenoside Rb1 by β-Glucosidase Extracted from L. buchneri URN103L

In this study, under optimal conditions (i.e., pH 5.0, 35 °C), the supernatant of the crude extract containing β-glucosidase from *L. buchneri* URN103L converted ginsenoside Rb1 to ginsenosides Rd and Rg3. After 14 days of hydrolysis, ginsenoside Rb1 was fully converted to ginsenoside Rd, and the transformation pathway was from ginsenoside Rb1 to ginsenoside Rd and then to Rg3, as shown by the results of TLC (Figure 6) and HPLC analysis (Figure 7). Clearly, Rb1 bioconversion increased with the incubation time, as indicated by the disappearance of Rb1 and the increasing intensity of the Rd band. Minor ginsenoside Rg3 was not detected by TLC. However, HPLC analysis (Figure 7) revealed the production of minor ginsenosides Rd and Rg3. In the structure of ginsenoside Rb1 (molecular weight: 1109.29), two glucose molecules are bound to 20-C of the protopanaxadiol ring. Among the two glucose molecules, one glucose molecule is hydrolyzed by β-glucosidase, and Rb1 is converted to Rd (molecular weight: 947.15); then, an additional glucose molecule is hydrolyzed by β-glucosidase, and Rd is converted to Rg3 (molecular weight: 785.0).

Although minor ginsenosides have various beneficial effects, only a small quantity is produced by enzymatic transformation in ginseng. Therefore, many studies have investigated enzymatic transformation methods using various beneficial microorganisms for conversion of major ginsenoside Rb1. It has been reported that various lactic acid bacteria isolated and identified from kimchi, a traditional Korean fermented food, can perform enzymatic biotransformation of ginsenoside. *Leuconostoc mensenteroides* WiKim19 and *Pediococcus pentosaceus* WiKim20 with β-glucosidase activity isolated from kimchi had a greater ability to transform ginsenoside Rb1 into ginsenosides Rg3 and Rg5 than the other strains [28]. In addition, *Lactobacillus plantarum* CRNB22 with β-glucosidase activity isolated from kimchi converted ginsenoside Rb1 into ginsenosides Rg3-s, Rg3-r, and Rg5 [29]. Depending on the enzyme type and experimental conditions, the final conversion of ginsenoside Rb1 was into Rd, Rg3, compound K, and other compounds [26]. Additionally, the crude enzyme extract from *Leuconostoc citreum* can convert ginsenoside Rb1 in the following sequence: Rb1→Rd→F2→compound K [27]. Consistently, β-glucosidase isolated from *D. anomala* YAE-1 fully converted ginsenoside Rb1 to ginsenoside Rd after 48 h of incubation [14].

### 3.7. Characteristics of the Fermentation of 20% Ginseng Root Solution by L. buchneri URN103L

The viable cell count of fermented ginseng roots is shown in Table 1. *L. buchneri* URN103L was inoculated at 3%. The initial viable cell count just after inoculation was approximately 6.04 ± 0.077 log colony-forming unit (CFU) mL^−1^. After 7 days of fermentation, the viable cell count increased to 8.27 ± 0.044 log CFU mL^−1^ and decreased by 7.39 ± 0.083 log CFU mL^−1^ over 14 days. The six types of LAB in fermented ginseng grew by approximately 9~11 log CFU mL^−1^ on the first day, and after 5 days, they decreased to 6.0~7.5 log CFU mL^−1^ [2]. Strain *L. buchneri* URN103L converted major ginsenoside Rb1 to minor ginsenosides Rd and Rg3 (epimer R/S), as shown in Figure 8. TLC analysis showed that the extent of conversion increased with time. Therefore, the growth time was increased to a 14-day period, such that there may be increased production of the enzyme β-glucosidase. Figure 9A shows the results of fermented Panax ginseng root. When *L. buchneri* URN103L in 20% ginseng root was incubated, ginsenosides Rb1 and Rd were metabolized into Rg3 after 3 days (Figure 9B). Subsequently, after 7 days, ginsenoside Rg3 (S/R) gradually appeared (Figure 9C). As shown in Figure 9D, the resultant minor ginsenosides Rg5, Rd, and Rg3 began to be detected, and their content increased over time. Among the 17 types of LAB exhibiting positive β-glucosidase activity from fermentation products of Korean plant foods, only 1 (i.e., URN103L) showed high hydrolytic activity on ginsenoside Rb1. The LAB strain was identified as *L. buchneri* URN103L, which showed 99% homology with *L. buchneri* NRRLB 30929. The optimum hydrolysis activity of the β-glucosidase enzyme on saponin occurred at 35 °C and pH 5.0. The identified strain URN103L showed higher β-glucosidase activities than any other strain. Since ginsenoside Rb1 itself is not a good growth substrate for *Lenti. buchneri*, ginsenoside Rb1 was converted using the crude enzyme in culture supernatants. However, since ginseng roots contain a variety of nutrients for the growth of *Lenti. buchneri*, ginsenoside Rb1 in the ginseng root suspension was converted by fermentation. This study confirms that the β-glucosidase of *L. buchneri* URN103L isolated from Korean fermented plant foods can effectively convert ginsenoside Rb1 into Rd and Rg3.

Ginsenoside Rg3, one of the saponins in American ginseng (*Panax quinquefolius* L. Araliaceae), has been shown to inhibit tumor growth. The most affected pathway of anti-human colorectal cancer activity is the Ephrin receptor pathway in HCT-116 human colorectal cancer cells [30]. In radiation therapy combined with surgery or chemotherapy, Rg3 was applied to inhibit the growth of cancer cells [31]. In addition, there are reports of research on the anticancer activity of ginsenoside Rg3 in breast cancer [32,33], colon cancer [34,35], gastric cancer [36], lung cancer [37], and liver cancer [38], among others [39]. Ginsenoside Rd is a biologically active component of ginseng that stimulates the proliferation of endogenous stem cells. Ginsenoside Rd was treated to evaluate its effect on gastrointestinal mucosal regeneration with an inflammatory bowel disease (IBD) rat model. [40]. We found that *Lentilactobacillus buchneri* URN103L with β-glucosidase activity, isolated from the fermentation of medicinal plants, has the potential to convert ginsenoside Rb1 into Rd and Rg3. In the future, we hope that various biological functions of ginsenosides, such as ginsenoside Rg3, will be revealed and play an important role in human health.

## 4. Conclusions

Among the 17 types of LAB exhibiting positive β-glucosidase activity from the fermentation products of Korean medicinal plants, only 1 isolate (i.e., URN103L) showed high hydrolytic activity on ginsenoside Rb1. The LAB strain was identified as *L. buchneri* URN103L, which showed 99% homology with *Lentilactobacillus buchneri* NRRLB 30929. The optimum hydrolysis activity of the β-glucosidase enzyme on saponins occurred at 35 °C and pH 5.0. The identified strain URN103L showed higher β-glucosidase activities than any other strain. This study confirms that the β-glucosidase of *L. buchneri* URN103L isolated from Korean fermented medicinal plants can effectively convert ginsenoside Rb1 into Rd and Rg3. The converted ginsenoside can be used in the food, cosmetic, and other health product industries.

## Figures and Tables

**Figure 1 foods-11-00529-f001:**
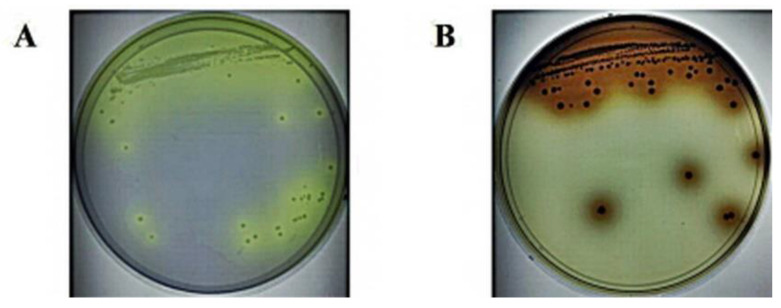
Screening of lactic acid bacteria with β-glucosidase activity from fermented plants: (**A**) single colony of strain URN103L in BCP agar, (**B**) positive reaction of strain. URN103L in esculin agar.

**Figure 2 foods-11-00529-f002:**
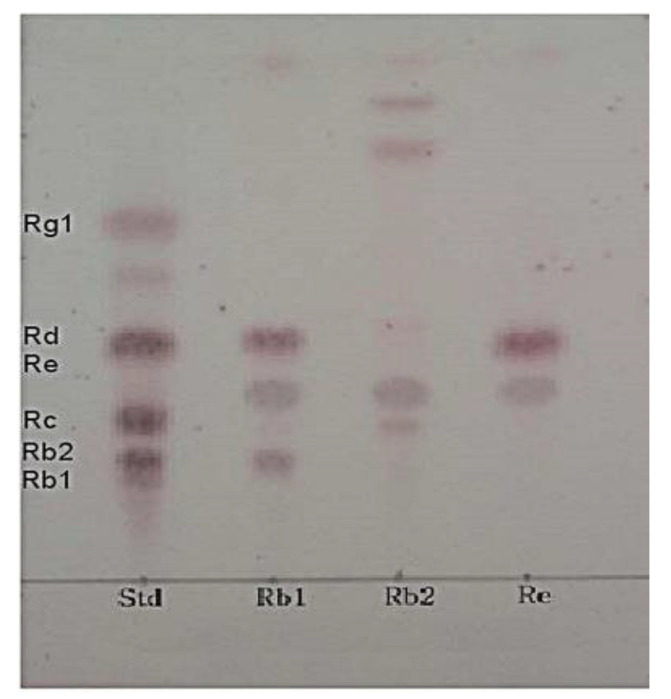
TLC analysis of ginsenosides hydrolyzed by strain URN103L in Rb1, Rb2, and Re for 7 days. Std: standard.

**Figure 3 foods-11-00529-f003:**
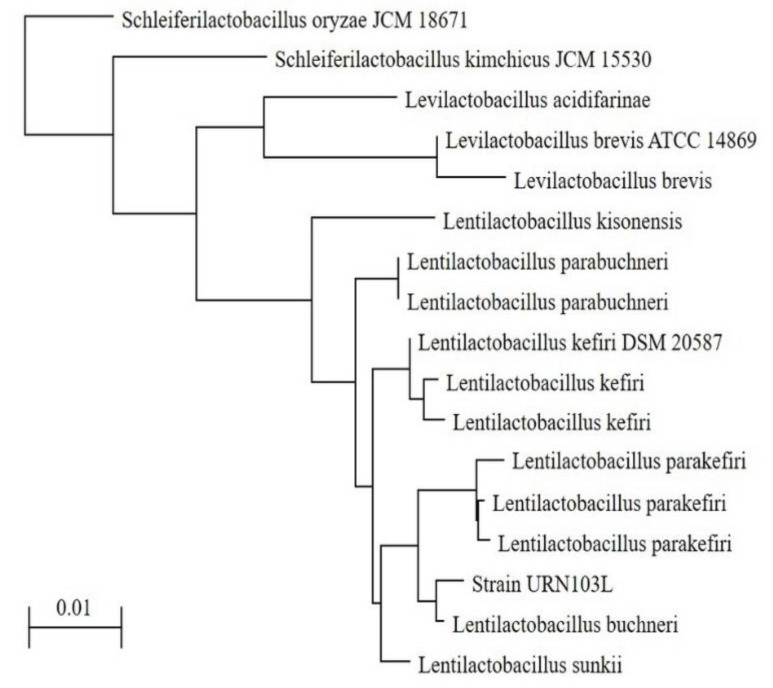
Phylogenetic tree based on 16S rDNA sequences. Phylogenetic relationships of strain URN103L with other *Lentilactobacillus* sp. are shown. Bar (0.01) is scale length.

**Figure 4 foods-11-00529-f004:**
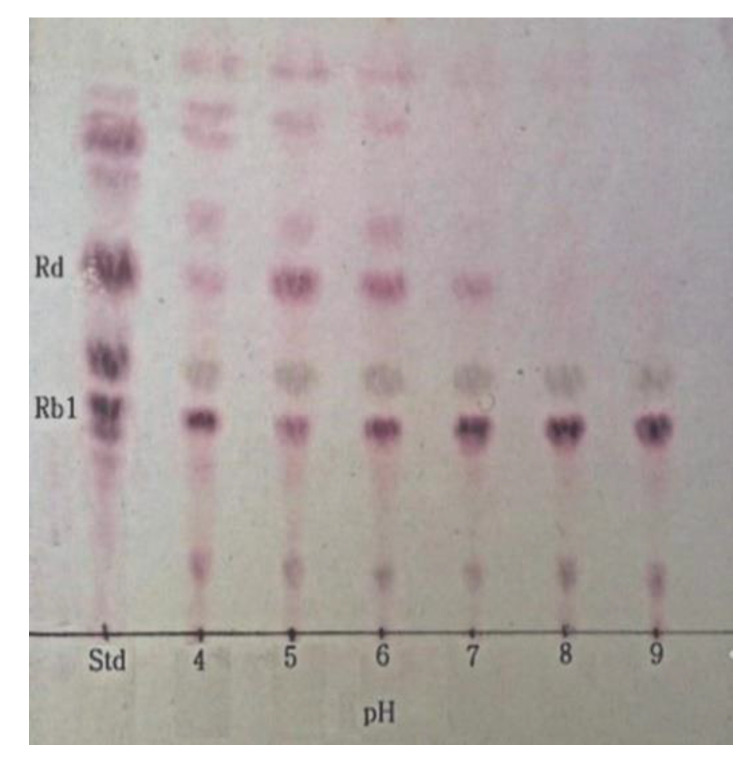
TLC analysis of ginsenoside Rb1 hydrolyzed by crude enzyme of isolated strain URN103L at different pHs (4.0, 5.0, 6.0, 7.0, 8.0, and 9.0) at 35 °C for 7 days.

**Figure 5 foods-11-00529-f005:**
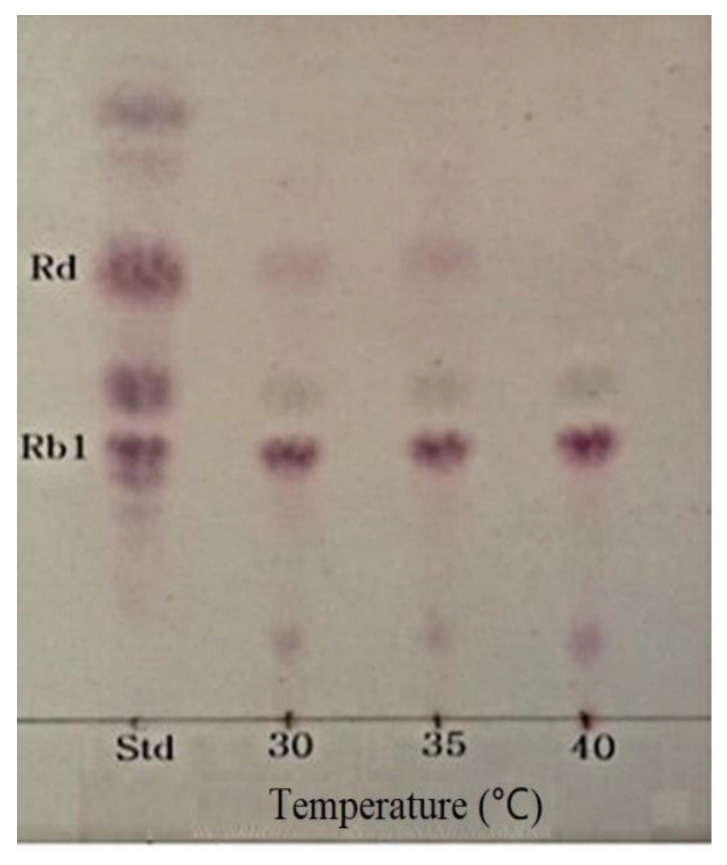
TLC analysis of ginsenoside Rb1 hydrolyzed by crude enzyme of isolated strain URN103L at different temperatures (30, 35, and 40 °C) at pH 7.0 for 7 days.

**Figure 6 foods-11-00529-f006:**
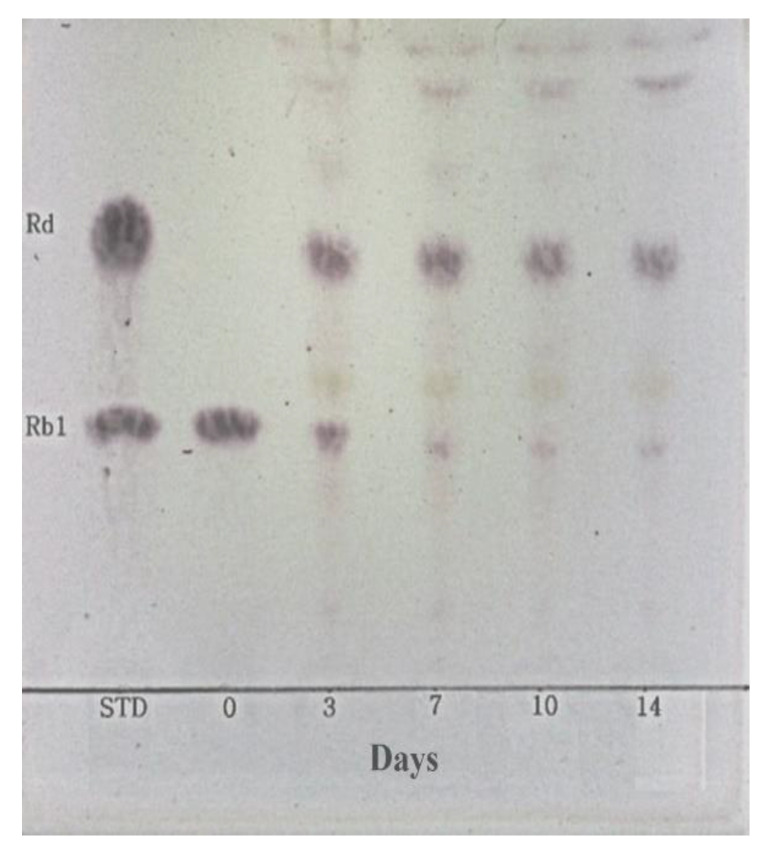
TLC analysis of ginsenoside Rb1 conversion by culture broth of *Lentilactobacillus buchneri* URN103L under the optimum condition for 14 days of hydrolysis. STD: standard.

**Figure 7 foods-11-00529-f007:**
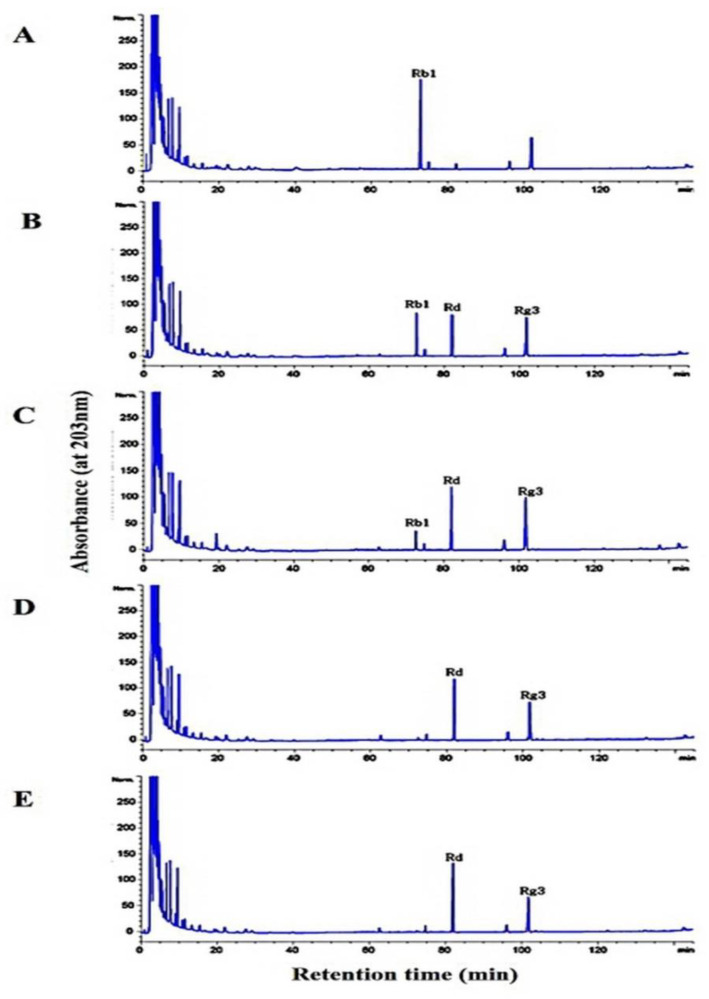
Conversion of ginsenoside Rb1 by crude enzyme from *Lactobacillus buchneri* URN103L: (**A**) 0 days; (**B**) 3 days; (**C**) 7 days; (**D**) 10 days; (**E**) 14 days.

**Figure 8 foods-11-00529-f008:**
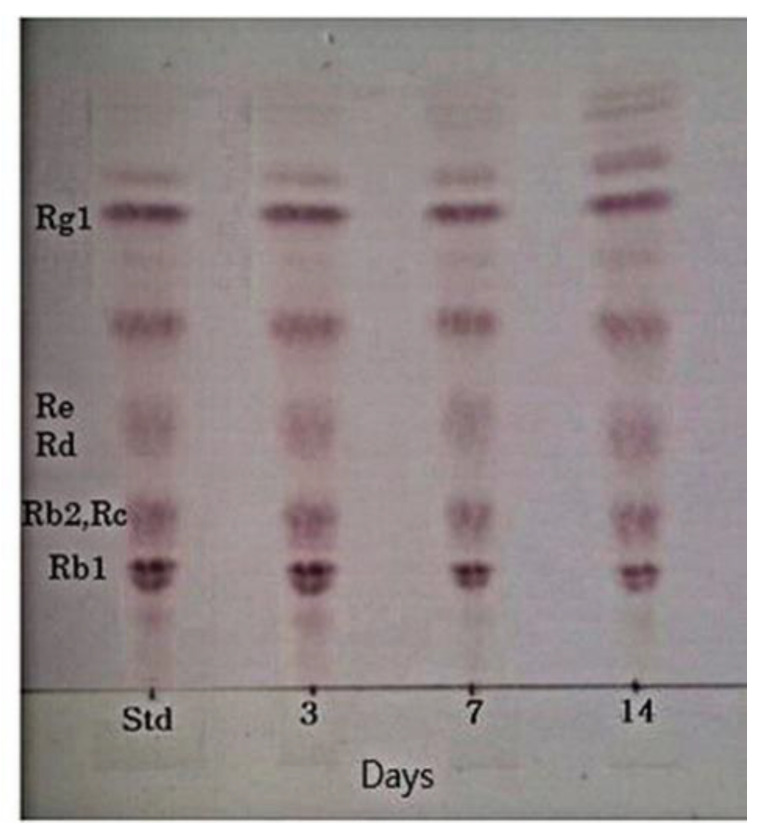
TLC analysis of ginsenoside root conversion by culture broth of *Lentilactobacillus buchneri* URN103L at optimum condition. Std: standard.

**Figure 9 foods-11-00529-f009:**
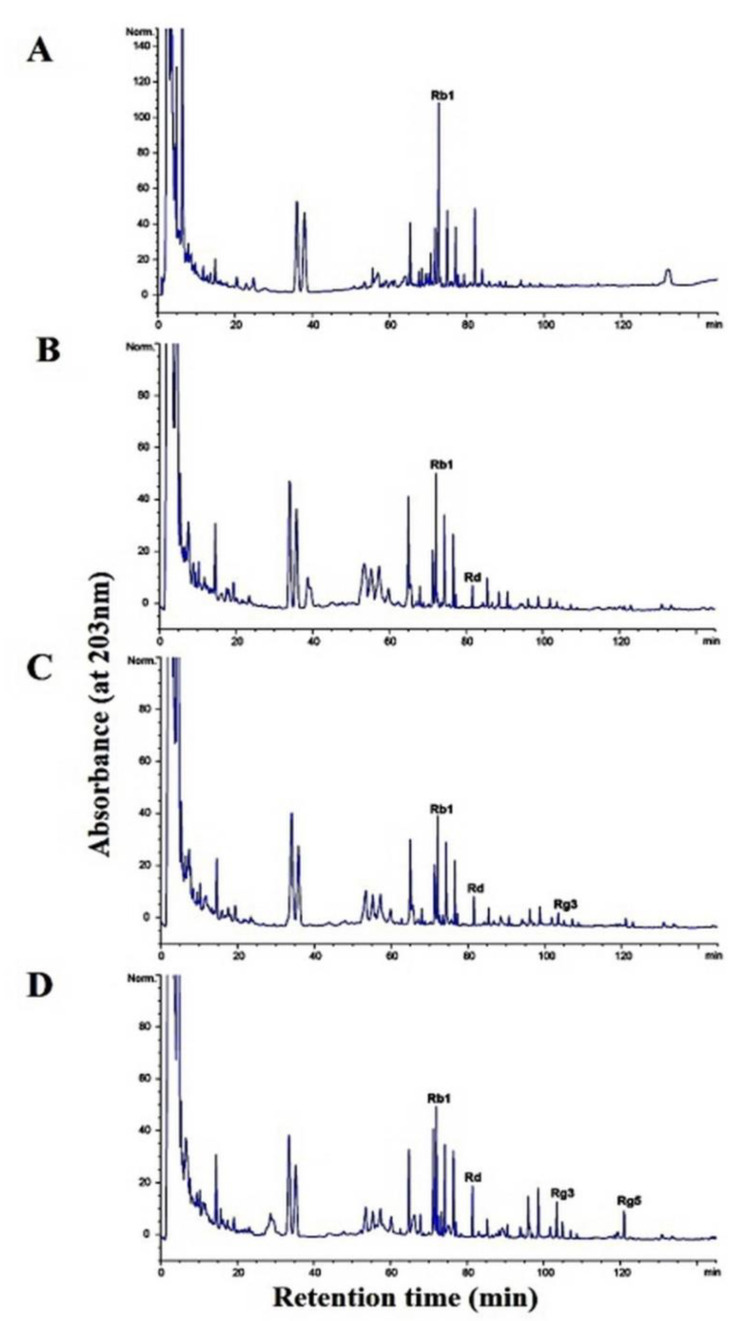
Ginsenoside conversion of ginseng root by *Lentilactobacillus buchineri* URN103L for fermentation: (**A**) Panax ginseng root; (**B**) 3 days; (**C**) 7 days; (**D**) 14 days.

**Table 1 foods-11-00529-t001:** Viable cell count of *Lentilactobacillus buchneri* URN103L after 14 days of fermentation in 20% ginseng root suspension.

Microorganism (Log CFU mL^−1^)	Incubation Time (Days)			
	0	3	7	14
*Lenti**lactobacillus buchneri* URN103L	6.04 ± 0.08 ^d^	7.88 ± 0.09 ^b^	8.27 ± 0.04 ^a^	7.39 ± 0.08 ^c^

Different superscript letters on each column represent significant differences (*p* < 0.05).

## Data Availability

The datasets used and analyzed in this study are available from the corresponding author on reasonable request.

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
