# Peer review of "Identification of β-Glucosidase Activity of Lentilactobacillus buchneri URN103L and Its Potential to Convert Ginsenoside Rb1 from Panax ginseng"

_foods, 2022, doi:10.3390/foods11040529_

Round 1

Reviewer 1 Report

General comments

The research is interesting. This type of research is missing and this is novum. I propose some improvements.

Detailed comments

  • Authors should apply new nomenclature of Lactobacillus in the whole manuscript, see: http://lactotax.embl.de/wuyts/lactotax/

Lactobacillus buchnerii is now Lentilactobacillus buchneri.

  • There are typing errors in the whole manuscript, e.g. double spaces, unnecessary capital letters, etc.
  • Panax ginseng” should be written in italics.
  • Authors should use “microbiota’ instead of “microflora” in relation to human microorganisms, as they have nothing in common with plants.
  • All abbreviations should be defined when used for the first time and then they should be used.
  • Methods: The methods are poorly described. The authors refer to literature for almost each method. It is confusing. Authors should describe the methodology in more details. Especially that there are always some modifications. Describe controls.
  • A part of the text in paragraphs 3.7 and 3.8 is repeated. See lines 263-379 and 285-301.
  • What is the biological activity of the converted ginsenoside in comparison to the parent one? Did authors made such type of research?
  • The Discussion is poor and should be extended. The differences in the biological activity of the substrate (Rb1) and the new ginsenosides (Rd and Rg3) should be discussed. Is it known from literature? How are the compounds different from each other? Do they have better application/health-promoting features than the initial one?
  • I understand the authors have incubated the compound with LAB supernatants. Did you try to culture the chemical directly with LAB / live cells? What could be the effect - it would be interesting. Maybe it should be discussed in the text.

Author Response

Response to Reviewer 1 Comments

General comments

The research is interesting. This type of research is missing and this is novum. I propose some improvements.

Detailed comments

  • Authors should apply new nomenclature of Lactobacillus in the whole manuscript, see: http://lactotax.embl.de/wuyts/lactotax/

Lactobacillus buchnerii is now Lentilactobacillus buchneri.

  • Lactobacillus buchnerii  →  Lentilactobacillus buchneri.
  • There are typing errors in the whole manuscript, e.g. double spaces, unnecessary capital letters, etc.
  • I have corrected everything you pointed out.
  • Panax ginseng” should be written in italics.
  • “Panax ginseng” → “Panax ginseng
  • Authors should use “microbiota’ instead of “microflora” in relation to human microorganisms, as they have nothing in common with plants.
  • Changed “microflora” to “microbiota’.
  • All abbreviations should be defined when used for the first time and then they should be used.
  • All abbreviations are used after definition.
  • Methods: The methods are poorly described. The authors refer to literature for almost each method. It is confusing. Authors should describe the methodology in more details. Especially that there are always some modifications. Describe controls.
  • If describe the methods in detail, the total repetition rate is over 30%. and the single repetition rate is over 5%. The detailed method have been uploaded to supplementary materials and methods.
  • A part of the text in paragraphs 3.7 and 3.8 is repeated. See lines 263-379 and 285-301.

The repetition of paragraphs 3.7 and 3.8 has been removed.

  • Line 285-301 removed.
  • What is the biological activity of the converted ginsenoside in comparison to the parent one? Did authors made such type of research?
  • Experiments on the biological activity of converted ginsenosides have not been performed.
  • The Discussion is poor and should be extended. The differences in the biological activity of the substrate (Rb1) and the new ginsenosides (Rd and Rg3) should be discussed. Is it known from literature? How are the compounds different from each other? Do they have better application/health-promoting features than the initial one?
  • The study of the biological activity of ginsenosides Rd and Rg3 was added to the discussion section. The structural change of ginsenoside Rb1 by β-glucosidase was added to the discussion section.

Line 312-349

  • Ginsenoside Rg3, one of the saponine in American ginseng (Panax quinquefolius Araliaceae), has been shown to inhibit tumor growth. It has demonstrated the downstream genes targeted by American ginseng extracts in HCT-116 human colorectal cancer cells. The most effected pathway of anti-human colorectal cancer activity is the Ephrin receptor pathway. This results suggest that Rg3 may exert effective anti-cancer activity through the Eph/ephrin pathway [30]. In radiation therapy combined with surgery or chemotherapy, Rg3 was applied to inhibit the growth of cancer cells. Rg3 treatment in γ-ray sensitized lung cancer cells suppresses NF (nuclear factor) -КB activity significantly, leading to the inhibition of tumor progression [31]. In addition, there are a reports of research on the anticancer activity of ginsenoside Rg3 is breast cancer [32, 33], colon cancer [34, 35], gastric cancer [36], lung cancer [37], liver cancer [38] and so on [39]. Ginsenoside Rd is a biologically active component of ginseng that stimulates the proliferation of endogenous stem cells. Ginsenoside Rd was treated to evaluate the effect on gastrointestinal mucosal regeneration with an inflamatory bowl disase (IBD) rat model. Ginsenoside Rd treatment, especially 20 mg/kg Ginsenoside Rd, significantly reduced indomethacin-induced damage compared with that in the control group. By measuring the mRNA and protein levels of the intestinal stem cell markers (Bmi, Msi-1, intestinal epithelial cell marker CDX-2, 5-bromo-2-deoxyuridine), Ginsenoside Rd could stimulate the proliferation and differentiation of endogenous intestinal stem cells in IBD model rats, leading to improved recovery of intestinal function [40]. Drugs targeting 26S proteasome as antitumor agents are considered to be important for cancer therapy. The ginsenoside Rd inhibited 52.9% the chymotrypsin-like activity of 26S proteasome. This result also suggest that ginsenoside Rd could function as a potential compound for cancer treatment and prevention [41]. We found that Lentilactobacillus buchneri URN103L with β-glucosidase activity isolated from fermentation of medicinal plant have potential to convert to ginsenoside Rb1 into Rd and Rg3. In the future, we hope that various biological functions of ginsenoside, such as ginsenoside Rg3, will be revealed and play an important role in human health.

Line 252-257                                                               

  • In the structure of ginsenoside Rb1 (molecular weight 1109.29), two glucose molecules are bound to 20-C of protopanaxadiol ring. Among the two glucose molecules, one glucose molecule is hydrolyzed by β-glucosidase and Rb1 is converted to Rd (molecular weight 947.15), and then additional glucose molecule is hydrolyzed by β-glucosidase and Rd is converted to Rg3 (molecular weight 785.0).
  •  
  • I understand the authors have incubated the compound with LAB supernatants. Did you try to culture the chemical directly with LAB / live cells? What could be the effect - it would be interesting. Maybe it should be discussed in the text.

   Line 305-309

  • Since ginsenoside Rb1 itself is not a good growth subtrate for buchneri, ginsenoside Rb1 was converted using the crude enzyme in culture supernatants. However, since ginseng roots contain a variety of nutrients for growth of Lenti. buchneri, ginsenoside Rb1 in the ginseng root suspension was converted by fermentation.

Reviewer 2 Report

The experiments  was well performed and interpreted. However, I have one suggestion. It is necessary for world wide researchers to submit 16S rRNA gene sequence to Gene bank such as NCBI, DDBJ, and so on. Please describe the accession number in the method 2.2.

Author Response

Response to Reviewer 2 Comments

The experiments was well performed and interpreted. However, I have one suggestion. It is necessary for world wide researchers to submit 16S rRNA gene sequence to Gene bank such as NCBI, DDBJ, and so on. Please describe the accession number in the method 2.2.

Line 79-80

  • The NCBI GeneBank registration number of strain URN103L is OM438138.1.

Reviewer 3 Report

Methodology.

Lines 66-67. Rewrite the paragraph “Lactobacillus strains were isolated from eight types of Korean 86 traditional fermented plant foods randomly collected from the Daejeon area”

Line 90. Correct the word “prpared”

Line 97. Correct the word “en-zymeextract”

Line 100. Correct the word “ofsample”

Results

Line 137. Many researchers have isolated L. buchneri with various biological activities … Please cite some of them.

Lines 215-228. What are the advantages or disadvantage of L. buchneri, URN103L having an optimum pH of 5 and 35 C, compare to others for β-glucosidase activity?

Line 256. The LAB with intracellular β-glucosidase 256 activity isolated from kimchi can only convert ginsenoside Rb1 into ginsenosides Rg3 and 257 Rg5. Is this an advantage or disadvantage? Add discussion.

Table 2. Add Standard deviation and report if there are statistically differences between means. And add the statistical analysis for doing this.

Author Response

Response to Reviewer 3 Comments

Methodology.

Lines 66-67. Rewrite the paragraph “Lactobacillus strains were isolated from eight types of Korean 86 traditional fermented plant foods randomly collected from the Daejeon area”

  • “86” → “removed”

Line 90. Correct the word “prpared”

  • “prpared” → “prepared”

Line 97. Correct the word “en-zyme extract”

  • “en-zyme extract” → “enzyme extract”

Line 100. Correct the word “ofsample”

  • “ofsample” → “of sample”

Results

Line 137. Many researchers have isolated L. buchneri with various biological activities … Please cite some of them.

Line 144-151

  • In this paper, line 143-150 already cited on buchneri.

Lines 215-228. What are the advantages or disadvantage of L. buchneri, URN103L having an optimum pH of 5 and 35 C, compare to others for β-glucosidase activity?

       Line 225-228

  • Based on TLC data analysis (Figure 4 and Figure 5), the optimum conversion conditions were determined to be pH 5 and 35 °C. Ginsenoside Rb1 which had the weakest band density at pH 5 and 35 °C, was converted to Rd and Rg3.

  • Line 256. The LAB with intracellular β-glucosidase 256 activity isolated from kimchi can only convert ginsenoside Rb1 into ginsenosides Rg3 and 257 Rg5. Is this an advantage or disadvantage? Add discussion.
  • There was an error in discussion, so I corrected it and added the contents.

Line 265-275

  • Although minor ginsenosides have various beneficial effects, only small quantity is produced by enzymatic transformation in ginseng. Therefore, many studies have investigated methods enzymatic transformation methods using various beneficial microorganisms for conversion of major ginsenosides Rb1. It has been reported that various lactic acid bacteria isolated and identified from Kimchi, a traditional Korean fermented foods, can perform enzymatic biotransformation of ginsenoside. The Leuconostoc mensenteroides WiKim19 and Pediococcus pentosaceus WiKim20) with β-glucosidase activity isolated from Kimchi had higher transformation ability of ginsenoside Rb1 into ginsenosides Rg3 and Rg5 than the other strains [28]. Also Lactobacillus plantarum CRNB22 with β-glucosidase activity isolated from Kimchi converted ginsenoside Rb1 into ginsenosides Rg3-s, Rg3-r and Rg5 [29].

Table 2. Add Standard deviation and report if there are statistically differences between means. And add the statistical analysis for doing this.

       Line 128-132

  • 9. Statistics analysis

All experiments were repeated in triplicates, and one way analysis of variance (ANOVA) and Duncan’s multiple range tests were conducted using SAS (Statistical Analysis System Institute, Version 9.4, Cary, NC) to measure significant differences (p < 0.05). Data are presented as mean ± standard error.

  • Table 2 → Table 1 changed. (Table 1 was deleted and described in sentences to lower the repetition rate)
  • CFU mL-1 → log CFU mL-1 (In order to sufficiency process statistics, we changed it to the log value)

Line 284-290

  • The initial viable cell count just after inoculation was approximately 04±0.077 log colony forming unit (CFU) mL-1. After 7 days of fermentation, the viable cell count increased to 8.27±0.044 log CFU mL-1 and decreased to 7.39±0.083 Log CFU mL-1 over 14 days. The six types of lactic acid bacteria (LAB) in fermented ginseng grew by approximately 9 ~ 11 log CFU mL-1 on the first day of fermentation, and after 5 days of fermentation, the viable cell count of LAB decreased to 6.0 ~ 7.5 log CFU mL-1 [2].

Table 1. Viable cell count of Lentiactobacillus buchneri URN103L for 14 days fermentation in 20% ginseng root suspension.

Microorganism (log CFU mL-1)

Incubation time (days)

0

3

7

14

Lentiactobacillus buchneri URN103L

6.04±0.08d

7.88±0.09b

8.27±0.04a

7.39±0.08c

Round 2

Reviewer 1 Report

Line 119: "Buchneri" write with a small letter.

The abbreviation for all new genera of Lactobacillus is "L.".

Author Response

Response to Reviewer 1 Comments

Comments and Suggestions for Authors

Thanks to your detailed review of this article, I would like to express my deep gratitude for the creation of this article as a higher-quality article.

The review was revised as follows.

Thank you very much.

Line 119: "Buchneri" write with a small letter.

  •  Lenti. BuchneriL. buchneri

The abbreviation for all new genera of Lactobacillus is "L.".

All Lenti. buchneri abbreviation in this manuscript have been revised to L. buchneri.

  •  Lenti. buchneri   L. buchneri
